# Circular Nucleic Acids Act as an Oncogenic MicroRNA Sponge to Inhibit Hepatocellular Carcinoma Progression

**DOI:** 10.3390/biomedicines13051171

**Published:** 2025-05-11

**Authors:** Qianyi Zhang, Pengcheng Sun, Guang Hu, Xuanyao Yu, Wen Zhang, Xuan Feng, Lan Yu, Pengfei Zhang

**Affiliations:** 1School of Life Sciences, Tianjin University, Tianjin 300072, China; zhangyishan0618@163.com; 2Institutes of Biomedical Sciences, School of Life Sciences, Inner Mongolia University, Hohhot 010070, China; yuxuanyao369@163.com (X.Y.); 111984066@imu.edu.cn (W.Z.); 14747346480@163.com (X.F.); 3Hangzhou Institute of Medicine, Chinese Academy of Sciences, Hangzhou 310018, China; hug1109@hnu.edu.cn; 4Department of Obstetrics and Gynecology, The Fourth Affiliated Hospital of Harbin Medical University, Harbin 150001, China; hmu601784@hrbmu.edu.cn; 5School of Biomedical Sciences, Hunan University, Changsha 410082, China; 6Institute of Basic Medicine, Inner Mongolia Academy of Medical Sciences, Hohhot 010010, China; 7Clinical Medical Research Center, Inner Mongolia Key Laboratory of Gene Regulation of the Metabolic Diseases, Inner Mongolia People’s Hospital, Hohhot 010010, China

**Keywords:** miRNA sponges, circular single-stranded nucleic acids, HCC, cancer therapy

## Abstract

**Background:** Aberrant expression of microRNAs in neoplastic lesions may serve as potential personalized therapeutic targets. To inhibit oncogenic microRNAs (oncomiRs) expression and restore tumor suppressor proteins, linear miRNA sponges have been developed, leading to several drugs in clinical trials. Despite their efficacy, chemically synthesized miRNA inhibitors face challenges with sustained inhibition and high production costs, hindering widespread clinical adoption. Additionally, single-stranded circular RNAs (circRNAs) act as miRNA sponges, enhancing protein expression and demonstrating stability and therapeutic potential in cancer treatment. Our approach involves the use of synthetic single-stranded circular nucleic acids, including circDNA and circRNA, to selectively target and inhibit a variety of aberrantly overexpressed oncomiRs in tumors. The objective of this strategy is to restore the expression levels of multiple tumor suppressor factors and to suppress the malignant progression of tumors. **Methods:** Our methodology comprises a two-step process. First, we identified tumor suppressor genes (TSGs) with abnormally low expression in hepatocellular carcinoma (HCC) tumor cells by transcriptomic analysis and targeted the upstream cancer miRNA clusters of these TSGs. Second, we designed and validated a fully complementary circDNA or circRNA construct, ligated by T4 DNA ligase or T4 RNA ligase, respectively, that specifically targets the sponge oncomiRs both in vitro and in vivo to inhibit the malignant progression of HCC. **Results:** CircNAs demonstrated superior, long-lasting therapeutic efficacy against HCC compared to inhibitors. Furthermore, we compared the immune effects in vivo of three different nucleic acid adsorption carriers, including commercial miRNA inhibitor, circDNA, and circRNA. We found that the miRNA inhibitor activates a more robust inflammatory response compared to circDNA and circRNA. **Conclusions:** These findings underscore the substantial therapeutic potential of circDNA in tumorigenesis and provide novel insights for the formulation of personalized treatment plans for malignant tumors, such as HCC.

## 1. Introduction

Liver cancer is the sixth most commonly diagnosed cancer worldwide. Hepatocellular carcinoma (HCC) accounts for more than 80% of liver cancer cases and ranks among the top three causes of cancer-related mortality in 46 countries [1]. Despite improvements in the prognosis of patients with advanced HCC, the overall 3-year survival rate remains well below 50% [2]. Treatment of early-stage HCC primarily involves surgical resection and liver transplantation. However, recurrence remains a significant challenge. Due to current limitations in prevention and early detection strategies, many patients are diagnosed only at advanced stages. Management of advanced HCC predominantly relies on systemic therapy, including chemotherapeutic agents and immune checkpoint inhibitors. However, existing treatment modalities are limited in efficacy and lack precision. Consequently, there is an urgent need to develop personalized therapeutic approaches for patients with HCC to improve clinical outcomes and quality of life.

MicroRNAs (miRNAs) are short, non-coding RNA transcripts that contain one or more hairpin structures [2]. MicroRNAs mediate post-transcriptional gene silencing by directly binding to the 3′ untranslated region (3′ UTR) or open reading frame (ORF) of specific target mRNAs, thereby inhibiting protein expression and promoting target mRNA degradation. These small RNAs play crucial roles in various aspects of cancer cell biology, including angiogenesis, epithelial–mesenchymal transition, metastasis, and drug resistance by targeting oncogenes and tumor suppressor genes (TSGs) [3,4,5,6]. Oncogenic miRNAs (OncomiRs) exert various effects on the occurrence and progression of HCC, exacerbating tumorigenesis [7]. These oncomiRs are also involved in regulating different signaling pathways within cancer cells. Thus, novel therapeutic strategies and approaches based on miRNA inhibitors urgently need to be developed for cancer.

In order to reduce the expression of oncomiRs and restore the expression of tumor suppressor proteins, various methods have been developed to block oncomiR function, including antisense oligonucleotide inhibitors and miRNA “sponges”. Consequently, several drugs targeting tumor-associated miRNAs have entered clinical trials, with some progressing to Phase I and II trials. One of the earliest miRNA-based molecules to enter clinical development is the locked nucleic acid (LNA) miravirsen, which is composed of a double-ring structure analog of RNA [8]. It exhibits high binding affinity for complementary RNA targets and high stability in blood and tissues in vivo. LNA oligonucleotides can mediate potent and specific inhibition of miRNA function both in vitro and in vivo. Joacim et al. designed LNA-antimiR oligonucleotides with sequences complementary to miR-122, resulting in specific antagonism of miRNA-122 in mice [9]. Currently, commercial microRNA inhibitors are primarily available in two forms: single-stranded RNA and single-stranded antisense oligonucleotides (ASOs). The linear single-stranded miRNA inhibitors are usually chemically modified agents designed to target specific miRNAs, competitively binding with mature miRNAs to reduce their gene silencing effects and thereby increase protein expression levels. Chemical modifications to antisense oligonucleotides (ASOs) can enhance hybridization affinity for target RNAs, resist nuclease degradation, or activate RNase H or other proteins involved in termination mechanisms [10]. Studies have shown that oligonucleotides with 2′ sugar modifications can enhance hybridization affinity for target RNAs and resist nuclease degradation, including modifications such as 2-O-methyl [11], 2-O-methoxyethyl [12], 2-fluoro [13], and LNA [14]. Phosphorothioate backbone modifications can activate RNase H or other proteins involved in termination mechanisms, resulting in effective inhibition. These modifications are crucial for in vivo delivery, as they enhance protein binding and delay plasma clearance. Additionally, the incorporation of thiophosphate groups improves the stability and pharmacokinetic properties of oligonucleotides, facilitating their therapeutic potential in various clinical applications [15]. Despite the efficacy of chemically synthesized miRNA inhibitors in targeting miRNAs, significant challenges persist in achieving sustained inhibition. Furthermore, the relatively high production costs associated with these inhibitors present additional barriers to their widespread adoption and clinical application [16].

As a complement to linear miRNA inhibitors, single-stranded circular RNAs (circRNAs) function as highly expressed miRNA sponges in vivo. These molecules exist in a circular conformation, in contrast to the typical linear structure of other RNA species. This unique circular configuration confers distinct properties and functionalities to circRNAs in their role as miRNA regulators [17]. For example, Song et al. [18] found that circSRY directly binds to miR-138-5p in spermatogonia. In vitro experiments showed that circSRY regulates H2AX mRNA by acting as a sponge for miR-138-5p. Endogenous circRNAs can function as molecular sponges for miRNAs in vivo, targeting miRNAs and thereby increasing protein expression. In another study, Stephen J. Meltzer et al. [19] designed synthetic circular RNAs in vitro containing five binding sites for miR-21, an oncomiR. These circRNAs exert their sponge effect by affecting the post-transcriptional modification of downstream genes, thereby suppressing gastric carcinoma cell proliferation. Synthetic circRNAs, which lack chemical modifications, exhibit good tolerance in biological systems and are more stable than linear RNAs in their miRNA sponge functions. Mimicking circRNA features, Yang et al. [20] developed an artificial single-stranded circDNA molecule containing multiple consecutive complementary sites for miR-9. Single-stranded circDNA effectively releases multiple co-silenced TSGs (*KLF17*, *CDH1*, and *LASS2*) by sequestering oncomiR-9, thereby upregulating their expression and inhibiting tumor progression and lung metastasis. Therefore, we plan to systematically study and characterize the functions and properties of circular single-stranded nucleic acids as microRNA sponges, with the aim of developing and enriching microRNA-based cancer therapeutic strategies.

In this study, we synthesized two different single-stranded circular nucleic acids (circNAs) to investigate their adsorption effects on oncomiRs. We employed T4 RNA ligase to generate circRNA [21,22] and utilized T4 DNA ligase to produce circDNA [23]. The designed and synthesized circDNA and circRNA effectively targeted oncomiRNAs and restored tumor suppressor gene expression in HCC. After optimizing the miRNA sponge ratio, we performed a comparative analysis of immune-activated inflammatory factor expression among three distinct nucleic acid carriers: miRNA inhibitors, circDNAs, and circRNAs. This comparison aimed to identify the most promising approach to improve therapeutic outcomes in HCC treatment.

## 2. Materials and Methods

### 2.1. Cell Culture

The transformed human liver epithelial-3 (THLE-3) cells were kindly provided by Huafeng Zhang (University of Science and Technology of China). The human hepatocellular carcinoma cell lines, HCCLM3 and HepG2, were obtained from the American Type Culture Collection (ATCC). The human hepatocellular carcinoma cell lines, PLC/PRF/5 (abbreviated as PLC) and Huh-7, were purchased from Pricella. All cell lines were authenticated by short tandem repeat (STR) profiling. HCCLM3, Huh-7, HepG2, and PLC cells were cultured in Dulbecco’s modified Eagle’s medium (DMEM) (Gibco, Brooklyn, NY, USA). THLE-3 cells were cultured in BEBM™ bronchial epithelial cell growth basal medium (LONZA, Walkersville, MD, USA). All cell lines were cultured in a medium supplemented with 10% fetal bovine serum (FBS) (Gibco, USA) and 1% penicillin/streptomycin (Gibco, USA) at 37 °C in a humidified atmosphere of 5% CO_2_.

### 2.2. RT-qPCR

For the extraction of total RNA from cells and tissues, the Trizol Universal Reagent (Tian Gen, Beijing, China) should be employed, following the instructions provided by the manufacturer. Real-time PCR was conducted using SYBR Green Fast qPCR Mix (Abclonal, Wuhan, China; RK21203) and PrimeScript™ RT Kit for synthesizing complementary DNA (Takara Bio, Kusatsu, Japan). The target gene was standardized as the housekeeping gene (β-actin) and displayed as 2^−ΔCt^. The primers utilized in qPCR are presented in Appendix A.

### 2.3. Absolute Quantification of RNA Expression

The standard curve method was employed to achieve absolute quantification of RNA expression. This method entailed plotting the Ct (cycle threshold) value against the copy number determined by the adjusted final concentration of DNA fragments, which was subsequently employed to construct a standard curve utilizing the full-length DNA fragment of the target cDNA. To quantify RNA expression, RNA was extracted from a fixed number of cells, and cDNA equivalents were prepared from 2000 cells per qPCR assay. The mean RNA copy number per cell was determined by calculating the Ct value of the sample within the linear range of the reference standard curve.

### 2.4. Western Blotting

The cultivation and processing of cells should be undertaken in accordance with the instructions provided. Prior to harvesting, the cells should be washed twice with pre-cooled PBS buffer. Total protein was extracted using RIPA lysis buffer (Beyotime, Shanghai, China), and protein concentration was normalized using a Lowry protein assay kit (Solar). The protein sample should be mixed with protein loading buffer and incubated at 95 °C for 10 min. The following antibodies should be used for protein blotting. Following an overnight incubation period with the primary antibody, the imprinted NC membrane was combined with goat anti-rabbit immunoglobulin G-horseradish peroxidase (HRP) secondary antibody (Jackson, West Grove, PA, USA; 611-624-215) and exposed for an appropriate duration using the Amersham ImageQuant 800 imaging system.

### 2.5. Extraction and Purification of circDNA

To commence the ligation process, we combine 80 picomoles of phosphorylated linear single-stranded DNA (Appendix A) with 80 picomoles of its corresponding primers in a reaction containing 10 microliters of 10X T4 DNA ligase buffer. This mixture is then subjected to a gradient annealing protocol (85 °C for 5 min; 80 °C for 2 min (−1 °C/cycle); 45 cycles; 12 °C ∞). The reaction is then treated with 5U of T4 DNA Ligase (Vazyme, Nanjing, China) and incubated at 16 °C for 16 h. Extract and remove enzymes using a mixed solution (phenol/chloroform/isostarch = 25:24:1, pH 7.4). Add 1/10 volume of 3M sodium acetate (pH 5.2) and 2.5 times the volume of ethanol to the extraction solution, then vortex and allow to stand at −20 °C for 2 h. After centrifugation at 4 °C (13,000 rpm, 35 min) and washing with 75% ethanol, circDNA was obtained. Finally, the precipitated DNA was dissolved in water.

### 2.6. In Vitro RNA Transcription, Circularization, and Purification

RNA transcripts were generated using the T7 High-Yield RNA Transcription Kit (Vazyme, China), with a double-stranded DNA vector (Appendix A) incorporating the T7 promoter as the template. For the production of circular RNA precursors, 2 μg of template DNA was first annealed and ligated to double-stranded DNA fragments via polymerase chain reaction (PCR). The resulting dsDNA was then subjected to in vitro transcription in a reaction mixture that included 2 μL of T7 RNA polymerase and a nucleotide blend consisting of ribonucleoside triphosphates (rATP, rCTP, rUTP) and guanosine monophosphate (GMP), each at a final concentration of 10 mM, along with riboguanosine triphosphate (rGTP) at 2 mM. The transcription reaction was maintained at 37 °C for a duration of 4 h. Following the in vitro transcription, the DNA template was digested using DNase I (Vazyme, China) at 37 °C for a period of 15 min to eliminate the template DNA. The resulting RNA was then purified through precipitation with lithium chloride (LiCl). The circularization of RNA was achieved using the T4 RNA ligase 1 cycle system, adhering to the protocol provided by the manufacturer. Specifically, 2 μg of linear RNA was incubated with T4 RNA ligase 1 (NEB, Ipswich, MA, USA) in a 20 μL volume at 37 °C for 3 h. Following this incubation period, the circular RNA was isolated from the reaction mixture by extraction with Trizol reagent, followed by purification involving the use of chloroform.

### 2.7. Serum Stability Assay

Serum stability of lssDNA/circDNA and lssRNA/circRNA constructs was evaluated by incubating samples (40 pmol) in DMEM supplemented with 10% FBS at 37 °C for 1, 4, 8, and 12 h. Degradation profiles were analyzed using 20% polyacrylamide gel electrophoresis (for DNA constructs: lssDNA scram and circDNA scram) or 2% formaldehyde-denaturing agarose gel electrophoresis (for RNA constructs: lssRNA scram and circRNA scram).

### 2.8. Annexin V FITC Assay

The procedure entailed detaching the cells with trypsin, followed by two washes with phosphate-buffered saline (PBS). In order to perform double staining of HCC cells, propidium iodide (PI) was used in conjunction with Annexin V-FITC, with the reagents obtained from the Annexin V/FITC Assay Kit (Beyotime Biotechnology, Shanghai, China; C1062L). Flow cytometric measurements were conducted on a flow cytometer, with the data subsequently collected and analyzed using FlowJo™ (version 10.10) software.

### 2.9. Transwell Migration Assay

In order to investigate the migratory capabilities of hepatocellular carcinoma (HCC) cells, a standard 24-well Transwell insert was utilized, equipped with transparent polyethylene terephthalate (PET) plastic membranes (Corning, Inc., Corning, NY, USA) featuring an 8-mm pore size.

### 2.10. Measurement of Lipid Peroxidation

Lipid peroxidation was quantified using the BODIPY 581/591 C11 probe (Beyotime Biotechnology). Briefly, cells cultured to 80% confluency in complete medium were washed twice with phosphate-buffered saline (PBS, pH 7.4) and detached using 0.25% trypsin-EDTA (Gibco, Brooklyn, NY, USA). Subsequently, the cells were incubated with 1:1000 diluted BODIPY 581/591 C11 in serum-free medium for 2 h at 37 °C under 5% CO_2_ with light protection, followed by three PBS washes (5 min each, 300× *g* centrifugation) to remove unbound probes. Flow cytometric analysis was performed on a with the FITC channel (excitation/emission: 488 nm/525 ± 20 nm) to detect probe oxidation. A minimum of 8000 single-cell events per sample were recorded, and data were analyzed using FlowJo™ software.

### 2.11. Detection of Intracellular Reactive Oxygen Species (ROS)

Intracellular ROS levels were measured using the 2′,7′-dichlorodihydrofluorescein diacetate (DCFH-DA) fluorescent probe (Beyotime Biotechnology, China). Cells in the logarithmic growth phase were seeded into 12-well plates at a density of 2 × 10^5^ cells/well and cultured for 24 h at 37 °C under 5% CO_2_ until reaching 80–90% confluency. The culture medium was replaced with serum-free medium containing 10 μM DCFH-DA, followed by incubation at 37 °C in the dark for 30 min. Uninternalized probes were removed by three gentle washes with PBS. Cells were then trypsinized, centrifuged at 300× *g* for 5 min, and resuspended in PBS for immediate analysis. Fluorescence intensity was quantified with the FITC channel (excitation/emission: 488 nm/525 ± 20 nm). A minimum of 8000 single-cell events per sample were acquired, and data were processed using FlowJo™ software.

### 2.12. Xenograft Tumor Experiment

Six-week-old male NOD/SCID mice (18–22 g) were housed under SPF conditions (22 ± 2 °C, 50 ± 10% humidity, 12 h light/dark cycle). Huh-7 cells in logarithmic growth phase were digested with 0.25% trypsin, centrifuged (1000 rpm, 5 min), and resuspended in a PBS/Matrigel mixture (1:1 ratio) at a density of 3 × 10^7^ cells/mL (maintained on ice to prevent matrix solidification). For tumor transplantation, 100 μL of the cell suspension (containing 3 × 10^6^ Huh-7 cells) was subcutaneously injected into the right dorsal flank of each mouse, with daily monitoring of injection sites. Tumor dimensions (length [L] and width [W]) were measured every 3 days using digital calipers, with volumes calculated as *V* = 0.5 × *L* × *W*^2^ to generate growth curves. On day 28, mice were euthanized by CO_2_ asphyxiation, followed by tumor dissection, weighing, photographic documentation, and fixation in 4% paraformaldehyde.

### 2.13. RayBio Mouse Inflammatory Antibody Array

BALB/c female mice were administered inhibitor mix, circDNA mix, and circRNA mix via tail vein injection for a duration of 6 h. Blood was collected from the eye sockets, and serum was extracted. The membrane of the inflammatory antibody array was sealed for one hour at room temperature, followed by overnight incubation with one milliliter of serum sample at 4 °C. Subsequently, the membrane was washed with 2 mL of 1× wash buffer I and then with 2 mL of 1× wash buffer II. Subsequently, each membrane was covered with 1 mL of diluted biotin-conjugated anti-cytokine antibody and incubated for 90 min at room temperature. Following the removal of the primary antibody via washing, the membrane was incubated with 2 mL of 1000-fold diluted HRP-conjugated streptavidin for a period of 2 h at room temperature. Subsequently, the HRP-conjugated streptavidin was removed, and the membrane was washed. The detection of immune complexes was accomplished through the use of a detection buffer, with visualization conducted through the application of a fluorescence imaging system. The relative levels of immunoreactivity were quantified using the ImageJ (version 1.53). All animal studies were approved by the Animal Research Ethics Committee of Hangzhou Institute of Medicine, Chinese Academy of Sciences (2023R-0125) and performed according to institutional guidelines.

## 3. Results

### 3.1. Screening of Oncogenic microRNAs and Target Tumor Suppressor Genes in HCC

The cancer gene expression profile from TCGA is utilized for the systematic identification of upregulated oncomiRs and downregulated mRNAs in HCC. A total of 7514 genes were found to be downregulated in tumor tissues compared to adjacent normal tissues. To identify the mRNAs most likely to be repressed by oncomiRs overactivation in HCC, it is necessary to compare this gene set with mRNAs potentially regulated by miRNAs, as documented in the miRTarBase and DIANA databases (Figure 1A–C). To determine the primary functions of the identified mRNAs, Gene Ontology (GO) and Kyoto Encyclopedia of Genes and Genomes (KEGG) pathway enrichment analyses were performed. The results revealed that pathways related to fatty acid degradation, extracellular matrix (ECM) receptor interaction, and focal adhesion were particularly enriched (Figure 1D). To further investigate the TSGs involved in the focal adhesion kinase (FAK) pathway, *ANGPTL1* and *SOCS3* were selected for further study. Furthermore, *ACACB* and *EHHADH* were selected to explore their potential roles in the regulation of fatty acid oxidation pathways. Subsequently, potential cancer-associated miRNA clusters upstream of *ANGPTL1*, *SOCS3*, *ACACB*, and *EHHADH* were predicted via integrated cross-analysis using multi-databases, including miRDB, TargetScan, RNAInter, and miRWalk (Figure 1E). To further confirm the inhibition of four mRNAs and the overactivation of related cancer miRNAs, human healthy liver cells (THLE-3) and four HCC cell lines (HepG2, HCCLM3, PLC, and Huh-7) were employed to evaluate the expression levels of the aforementioned mRNAs and related miRNAs. The expression levels of *ANGPTL1*, *SOCS3*, *ACACB*, and *EHHADH* are all repressed at both mRNA and protein levels in HCC cells (Figure 1F,G). Among the potential regulatory oncomiRs, has-miR-9-5p, 182-5p (targeting *ANGPTL1*), 19a-3p (targeting *SOCS3*), 452-3p (targeting *ACACB*), and 589-3p (targeting *EHHADH*) exhibited increased expression in HCC cells (Figure 1H and Appendix A). The above data suggest that oncomiRs can be targeted and adsorbed to restore the expression profiles of tumor suppressor gene mRNAs, potentially offering therapeutic benefits. The next step is to target the pathways of tumor migration and fatty acid degradation. CircNAs will be further used to enhance the expression of *ANGPTL1*, *SOCS3*, *ACACB*, and *EHHADH*, and evaluate their effects at the cellular level. We will compare their efficacy with commercial inhibitors in terms of restoring mRNA and protein levels, as well as differences in therapeutic outcomes.

### 3.2. CircNAs Act as Competitive Inhibitors of OncomiRs, Derepressing ANGPTL1 and SOCS3 Expression to Suppress HCC Cell Proliferation and Migration While Promoting Apoptosis

As an initial attempt, we focused on the ANGPTL1 and SOCS3 genes to investigate the feasibility of developing a circNAs strategy to target apoptosis and inhibit the malignant progression of HCC cells. To ascertain the targeting effect of oncomiRs on each downstream mRNA, commercially available linear microRNA inhibitors (miRNA inhibitors), in the form of 21–23 nt 2′-O-methylated RNA oligonucleotides, were transfected into HCC cells. As anticipated, in both Huh-7 and PLC cells, miRNA inhibitors effectively replenished the mRNA and protein levels of TSGs (Figure 2A,D and Appendix A). In vivo, circRNA functions as a sponge for miRNAs. Compared to linear RNA, circular RNA is characterized by three key features: covalent closure, lack of a 5′ cap, and absence of a 3′ polyadenylation tail. These features confer remarkable stability to circRNA both in vivo and in vitro. To investigate the analogous functions and properties of closed-loop single-stranded RNA and DNA, we designed circNA constructs, including circDNA and circRNA (Appendix A). To evaluate the biostability of circular nucleic acids under physiological conditions, serum stability assays were conducted, revealing distinct degradation patterns between circular and linear structures. While linear single-stranded DNA (lssDNA scram) exhibited significant degradation within 8 h, circular DNA (circDNA scram) maintained structural integrity for over 12 h (Appendix A). In contrast, both linear single-stranded RNA (lssRNA scram) and circular RNA (circRNA) demonstrated substantial degradation after 8 h of serum exposure (Appendix A). These findings demonstrate that circDNA is a more stable nucleic acid molecular sponge than circRNA in physiological environments.

We first examined the ability of circDNA and circRNA to rescue ANGPTL1 and SOCS3 mRNA expression by sequestering corresponding upstream miRNA clusters (miR-9-5p, miR-182-5p, miR-19a-3p) in the FAK pathway, as identified by KEGG analysis in HCC cells. Both circDNA and circRNA significantly upregulated the expression of ANGPTL1 and SOCS3 in Huh-7 cells and PLC cells (Figure 2B,C,E,F and Appendix A). To validate the consistency across control groups, all control conditions—including mock untreated, inhibitor negative control (inhibitor-NC), scrambled circDNA (ciD-scram), and scrambled circRNA (ciR-scram)—were systematically evaluated. The results demonstrated equivalent mRNA and protein expression profiles of ANGPTL1 and SOCS3 between the untreated and scrambled control groups (Appendix A).

Given the significant enhancement of tumorigenicity, cell motility, and angiogenesis by ANGPTL1 and SOCS3 in the FAK pathway of HCC cells, both in vitro and in vivo, we sought to elucidate the mechanisms by which circDNA and circRNA influence HCC cell migration. It was demonstrated that ciD-9-5p, ciD-182-5p, and ciD-19a-3p, as well as ciR-9-5p, ciR-182-5p, and ciR-19a-3p, effectively inhibited HCC cell migration (Figure 2G–L). It was demonstrated that these designed and synthesized circNAs, akin to miRNA inhibitors, can target and adsorb oncomiRs, restore tumor suppressor gene expression levels, and inhibit HCC migration at the cellular level.

Next, we aim to investigate whether circNAs offer enhanced therapeutic potential for HCC. We examined changes in apoptosis and proliferation of HCC cells following transfection with circNAs and inhibitors. Previous studies have shown that ANGPTL1 promoted apoptosis by suppressing the STAT3/Bcl-2-mediated anti-apoptotic pathway and reduced cell migration and invasion [24,25,26,27]. SOCS3 is a member of the SOCS family, which serves to regulate negative feedback in the JAK2/STAT3 signaling pathway [28,29,30]. SOCS3 plays a crucial role in the anti-tumor effects of the pathway by promoting cell apoptosis through inhibition of STAT3 phosphorylation and subsequent downstream factors [31]. We found that ciD-9-5p, ciD-182-5p, ciD-19a-3p, and ciR-9-5p, ciR-182-5p, and ciR-19a-3p promoted apoptosis of HCC cells (Figure 3A–F) and exhibited inhibitory effects on HCC cell proliferation (Figure 3G,H). Quantification of apoptosis indicated that the circDNA- or circRNA-transfected group elicited a more pronounced therapeutic response than the inhibitor-transfected group, despite achieving comparable levels of apoptosis in HCC cells. To assess the durability of therapeutic effects, we performed longitudinal cell viability assays following circNAs and miRNA inhibitor treatments. These experiments demonstrated that the circNA-treated groups maintained significant anti-proliferative effects throughout the 10-day observation period. In contrast, the inhibitory activity of miRNA inhibitors was markedly diminished by Day 7. This temporal divergence underscores the sustained therapeutic efficacy of circNAs in vitro, further supporting their potential as durable epigenetic modulators in HCC intervention (Appendix A). Based on these findings, we demonstrate that circNAs restore tumor suppressor gene (TSGs) expression levels, target migration-related pathways in HCC, and exhibit sustained therapeutic efficacy at the cellular level.

### 3.3. CircNAs Act as Competitive Inhibitors of OncomiRs, Derepressing ACACB and EHHADH Expression to Suppress Cell Proliferation While Promoting Ferroptosis in HCC Cells

Furthermore, the *ACACB* and *EHHADH* genes were targeted to investigate the feasibility of single-stranded nucleic acid drugs for specific regulation of apoptosis to inhibit the malignant progression of HCC cells. As mentioned above, we first confirmed the targeting effect of miR-452-3p and miR-589-3p on *ACACB* and *EHHADH* in cancer cell lines using miRNA inhibitors. Consistent with our hypotheses, in both Huh-7 and PLC cells, these inhibitors significantly increased the mRNA and protein expression levels of the putative TSGs (Figure 4A,D and Appendix A). Subsequently, we engineered individual circDNA and circRNA constructs. The aim of this study was to determine the ability of circDNA and circRNA to rescue the mRNA expression of *ACACB* and *EHHADH* by sequestering the corresponding upstream miRNA clusters (miR-452-3p, miR-589-3p) in the fatty acid degradation pathway, as analyzed by KEGG in HCC cells. As expected, in both Huh-7 and PLC cells, the use of circDNA and circRNA resulted in a significant upregulation of *ACACB* and *EHHADH* expression at both the mRNA and protein levels (Figure 4B,C,E,F and Appendix A). To confirm the uniformity of control groups, a comprehensive assessment was conducted across all control conditions, including mock untreated samples, inhibitor negative controls (inhibitor-NC), scrambled circDNA (ciD-scram), and scrambled circRNA (ciR-scram). Analyses revealed comparable mRNA and protein expression levels of ACACB and EHHADH between untreated and scrambled control cohorts (Appendix A).

Previous research has demonstrated that *ACACB* is an enzyme that catalyzes the carboxylation of acetyl-CoA to malonyl-CoA, thereby promoting fatty acid synthesis and inhibiting fatty acid oxidation. During periods of energy stress, AMP-activated protein kinase (AMPK) inhibits fatty acid synthesis by phosphorylating and suppressing *ACACB*, which plays a crucial role in regulating the balance between fatty acid synthesis and oxidation [32,33]. This process is known to mediate the activation of ferroptosis. *EHHADH* plays a crucial role in fatty acid metabolism in HCC, and its downregulation is associated with the induction of ferroptosis [34]. Subsequently, we validated the effects of *ACACB* and *EHHADH* upregulation on lipid peroxidation and reactive oxygen species (ROS) levels in HCC cells. The results showed that the upregulation of *ACACB* and *EHHADH* in HCC cells led to an increase in lipid peroxidation and ROS levels, thereby promoting the occurrence of ferroptosis (Figure 4G–J). After transfection of circDNA into HCC cells, a marked downregulation of the ferroptosis marker gene GPX4 [35] was observed, accompanied by an upregulation of ACSL4 [36] within the AMPK signaling pathway (Figure 4K). The above results indicate that our designed circNAs and miRNA inhibitors can effectively target and adsorb oncomiRs, restore tumor suppressor gene expression levels, and increase HCC ferroptosis at the cellular level. Next, to verify whether circNAs targeting the fatty acid degradation pathway have therapeutic effects on HCC, our research results interestingly showed that transfection with miRNA inhibitors and circNAs targeting miR-452-3p and miR-589-3p effectively hindered the proliferation of HCC cells (Figure 4L,M). In contrast, treatment with circRNA did not significantly affect the proliferation of HCC cells. These data suggest that the upregulation of TSGs by circDNA transfection may suppress the tumorigenesis and progression of HCC, potentially offering a novel therapeutic strategy.

### 3.4. Manipulating Transcriptional Networks via OncomiR Sponge Molecules to Intensify Hepatocellular Carcinoma Regression and Immune Efficacy

In order to achieve a superior therapeutic effect, it was necessary to determine the ratio of sponge sites for five miRNAs. Taking circDNA as an example, different concentrations (0–160 pmol) of circDNA were transfected into HCC cells individually to ascertain the extent of upregulation in the expression levels of several oncomiRs in HCC cells (Appendix A). The results demonstrated that 40 pmol of ciD-9-5p and ciD-589-3p and 20 pm of ciD-182-5p, ciD-19a-3p, and ciD-452-3p had a pronounced effect on restoring the mRNA expression of *ANGPTL1*, *SOCS3*, *ACACB*, and *EHHADH*, respectively, in HCC cells. Based on this observation, five different ratios of circDNA mixtures were designed and subsequently confirmed by cell proliferation experiments, which displayed that ciD-mix-1 was the most efficacious combination for inducing apoptosis (Appendix A). Furthermore, our observations revealed that both the circDNA and circRNA mixtures potently augmented the mRNA expression levels of *ANGPTL1*, *SOCS3*, *ACACB*, and *EHHADH* (Appendix A). The above experiments indicate that we identified the optimal ratio of circNAs through drug set design, which we then applied to subsequent transcriptome sequencing and the assessment of the immunogenicity of different circNAs in mice.

To characterize the transcriptional reprogramming elicited by the inhibitor-mix, circDNA-mix, and circRNA-mix in HCC cells, we transfected the HCC cells with a meticulously optimized ratio of these mixtures, followed by extensive transcriptomic profiling (Figure 5A). Utilizing Gene Set Enrichment Analysis (GSEA), we observed that the inhibitor, circDNA, and circRNA experimental groups exhibited increased expression of genes predominantly associated with focal adhesion and fat digestion and absorption pathways compared to the control group (Figure 5B–D), which reflects our previous results (Figure 2G–L and Figure 4G–J). In addition to comparing treatment outcomes, we observed that the total genes in the inhibitor-mix group were predominantly enriched in neutrophil extracellular trap formation and cytokine–cytokine receptor interaction as compared to both the circDNA- and circRNA-mix groups (Figure 5E,F). This suggests a stronger activation of immune factors and a heightened inflammatory response. The circDNA-mix group exhibited a more pronounced enrichment of total genes within pathways involving cytokine-cytokine receptor interactions, the complement and coagulation cascade, and neutrophil extracellular trap formation when compared to the circRNA-mix group (Figure 5G). These results indicate that, compared to the three circNAs, the inhibitor-mix group induced relatively stronger immunogenic responses in HCC cells than both circDNA- and circRNA-mix groups, potentially translating into enhanced therapeutic effects in vivo. Furthermore, compared to the circRNA-mix group, the circDNA-mix group is more enriched in the cytokine activation pathway, which may lead to improved therapeutic outcomes.

The transcriptomic landscape revealed distinct regulatory hierarchies among therapeutic formulations. Following transfection with circNAs and inhibitor mixtures, transcriptomic analysis of HCC cells demonstrated activation of canonical tumor suppressor genes (TSGs), including *ANGPTL1*, *SOCS3*, *ACACB*, *EHHADH*, and *TP53* (Appendix A). Notably, circDNA-based combination therapy induced broader transcriptional modulation compared to controls, simultaneously suppressing emerging HCC-associated oncogenic markers such as *KIF*, *ABL1*, *ANXA2*, and *NCAPG* (Appendix A). Parallel observations in the circRNA combination cohort revealed sustained downregulation of key classical oncogenes (*PIK3CA*) and novel tumor-associated genes (*ANXA2*, *KIF2A*, *NCAPG2*, PAIP1) (Appendix A). Comparative analysis indicated that both circular nucleic acid formulations (circDNA/circRNA) induced more extensive transcriptional reprogramming than the inhibitor mixture alone, potentially through multi-target regulatory mechanisms.

To investigate the sustained therapeutic effects of circNAs in vivo, we administered optimized circNA formulations (inhibitor-mix, circDNA-mix, and circRNA-mix) via intratumoral injection in murine CDX models of HCC (Figure 5H). Quantitative evaluation of anti-tumor efficacy over a 21-day intervention period revealed a significant reduction in tumor volume in the treatment groups compared to controls (Figure 5I,J). Post-euthanasia tumor dissection and precise electronic balance measurements further confirmed statistically significant reductions in tumor weight in circNA-treated cohorts (Figure 5K). These findings demonstrate that circNAs effectively suppress HCC progression within the physiological tumor microenvironment, exerting durable and potent growth inhibition. By integrating transcriptomic profiling with the established animal model, this study systematically deciphers the dynamic molecular networks mediated by circNAs, providing both theoretical foundations and translational frameworks for the development of circNA-based nucleic acid interference therapies.

### 3.5. OncomiR Sponge Molecular Drugs Elicit Inflammatory Factor Activation

To evaluate the rapid innate immune activation triggered by circNAs and 2′O-methylated RNA inhibitors, serum immunogenicity was assessed at 6 h post-administration based on the characteristic time window of pattern recognition receptor-mediated cytokine production [37]. Given the ability of both single- and double-stranded RNA and DNA to induce innate immune responses in the organism, this study aims to delineate the immunomodulatory effects of synthesized circDNA, circRNA, and commercially available 2′O-methylated RNA inhibitors when administered in vivo. For this purpose, we prepared mixtures of circDNA, circRNA, and 2′O-methylated miRNA inhibitors in an optimized ratio for intravenous administration to mice. After 6 h, we assessed the inflammatory mediators induced by different miRNA sponge molecular drugs. The expression levels of 40 different inflammatory protein candidates in mouse serum were detected and compared through antibody array technology (Figure 6A). The results showed that the inhibitor mixture significantly stimulated the production of pro-inflammatory cytokines, including IL1α, IL3, IL6, IL9, and IL12, along with the induction of inflammatory chemokines such as MIP-1-α, Lymphactin, Eotaxin-2, SDF-1, MCP-1, TCA-3, LIX, and KC. CircDNAs promoted the activation of the inflammatory chemokine Eotaxin-2, the pro-inflammatory factor IL12, and the anti-inflammatory factor G-CSF, while circRNAs were associated with the activation of the pro-inflammatory cytokine IL9 and the inflammatory chemokines SDF-1 and KC (Figure 6B). A comparative analysis of the immunostimulatory effects induced by the 2′O-methylated miRNA inhibitor mix versus the circDNA-mix and circRNA-mix reveals that the inhibitor mix triggers a more pronounced activation of pro-inflammatory cytokines and chemokines, signifying its capacity to evoke a more vigorous innate immune response (Figure 6C). The quantitative fold changes of these inflammatory cytokines are presented in Appendix A.

Inflammatory cytokines, including IL-1α [38], IL-3 [39], IL-9 [40], and IL-12 [41], are pivotal in amplifying the immune response, particularly in immune surveillance and the clearance of tumor cells, by facilitating the proliferation, differentiation, and activation of immune cells. Nevertheless, the production of pro-inflammatory cytokines is not without consequence, as it may precipitate unwarranted inflammation or autoimmune phenomena. Subsequent to these observations, we conducted a KEGG pathway analysis on the inflammatory cytokine dataset, revealing that the inhibitor-mix and circRNA injection groups exhibited differential gene expression enrichment in pathways associated with cell proliferation, such as the JAK/STAT signaling pathway, compared to the PBS control group (Figure 6D). Prior research has elucidated that IL-6 engages the JAK-STAT3 pathway, contributing to a multitude of biological processes, including inflammation and angiogenesis, and it may foster tumor progression through involvement in cancer cell phenotypes such as anti-apoptosis, proliferation, and angiogenesis. Therapeutic agents targeting the IL-6/STAT3 pathway have shown promise in the treatment of non-small cell lung cancer and are poised to emerge as efficacious interventions in this domain [42]. Consequently, the role of IL-6 as a pro-inflammatory cytokine in tumor immunity may not be as deleterious in tumor promotion as conventionally perceived. Additionally, the activation of IL-3, as evidenced by our findings, indicates that the inhibitor injection group is significantly enriched in hematopoietic pathways (Figure 6D). Typically, IL-3 functions as a colony-stimulating factor (CSF) for bone marrow cells, thereby enhancing hematopoiesis. It stimulates the proliferation of diverse bone marrow cell lineages, including granulocytes, macrophages, megakaryocytes, eosinophils, basophils, mast cells, and erythroblasts, thereby enhancing immune activity and bolstering tumor immunity [43]. Hence, the inflammatory mediators elicited by nucleic acid therapeutics in murine models may embody a double-edged sword, simultaneously activating immune responses to combat tumors while potentially exacerbating tumor development.

## 4. Discussion

Conventional cancer therapies often fail to achieve their intended therapeutic outcomes, primarily due to their lack of specificity and the propensity for tumor recurrence. Our strategy of simultaneously silencing multiple TSGs offers the potential to enhance the efficacy of tumor therapy. In this study, we utilized transcriptomic analysis to identify TSGs and their corresponding oncomiRs involved in critical pathways, including tumor cell migration, apoptosis, and fatty acid degradation. We developed a novel approach by synthesizing covalently linked circDNA and circRNA with sponge-like functions, which, unlike miRNA inhibitors, avoids potential metabolite accumulation in the body. This innovative strategy may provide a targeted therapeutic avenue for cancer treatment (Figure 7).

This diagram would visually emphasize the complete complementarity of binding sites, the multi-targeting ability of circDNAs, and the differential immune responses elicited by circDNAs, circRNAs, and miRNA inhibitors in vivo. Use color coding and detailed molecular representations to highlight these key aspects and their implications for anti-tumor activity.

In HCC cells, deploying inhibitors and circDNA/circRNA constructs containing multiple oncomiRs binding motifs facilitates effective oncomiR sequestration. This interaction competitively reduces the translational repression imposed by oncomiRs on specific TSGs, including *ANGPTL1*, *SOCS3*, *ACACB*, and *EHHADH*. The pro-apoptotic and anti-migratory properties of ANGPTL1 and SOCS3 contribute to the inhibition of HCC cell malignancy. Furthermore, ACACB and EHHADH promote lipid peroxidation and ferroptosis, processes that impede HCC onset and progression. The sequence versatility of circDNA and circRNA, functioning as high-capacity molecular sponges targeting cancer-associated miRNA clusters, represents a promising novel therapeutic modality for HCC treatment.

CircRNAs constitute a class of endogenous long non-coding RNAs characterized by their covalently closed loop structure, which lacks conventional 5′-3′ polarity and poly (A) tail. These molecules can function as microRNA (miRNA) sponges, interact with RNA-binding proteins, modulate mRNA stability, and participate in gene transcription and protein translation processes [17,44]. Due to their conservation across species, abundance, and tissue-specific expression patterns, circRNAs are emerging as promising novel biomarkers for various diseases, particularly cancer. Accumulating evidence suggests that circRNAs serve as a critical structural platform for developing miRNA-targeted therapeutic approaches. However, upon miRNA binding, circRNAs recruit Argonaute (Ago) proteins, which promote miRNA degradation but also simultaneously induce circRNA cleavage, leading to its degradation and reduced anti-miRNA efficacy [45].

The engineered circDNA constructs utilize RNase H activity to degrade miRNAs while remaining resistant to RNase H digestion themselves, indicating potential for sustainable and reusable applications. These characteristics position circRNA and circDNA as promising candidates for the targeted inhibition of oncomiRs. Our findings have unveiled a promising and valuable avenue of research for the treatment of HCC and other malignancies.

The capacity of the immune system to identify exogenous sequences within the complex matrix of self-nucleic acids constitutes a pivotal characteristic of its defensive strategy. In alignment with the editorial standards of *Nature* publications, the cascade of cellular responses to viral challenge in mammals is initiated by exogenous RNA, triggering defense mechanisms via interferon-mediated and NF-κB-dependent signaling pathways [46,47]. Toll-like receptors 3 (TLR3) and 7 (TLR7) serve as sentinels within endosomes, alert to the detection of double-stranded RNA (dsRNA), while TLR7 and TLR8 monitor for the presence of single-stranded RNA (ssRNA). As for DNA, cells can recognize exogenous DNA through a cytoplasmic sensor such as cGAS. cGAS interacts with the stimulator of interferon genes (STING), facilitating the production of type I IFNs upon invasive cytosolic DNA exposure [48,49]. The STING pathway, in conjunction with IRF3 and the pro-apoptotic factor BCL-2-associated X protein (BAX), responds to dsDNA by initiating apoptosis. DNA sensors, including breast cancer type 1 susceptibility protein (BRCA1), DNA-dependent protein kinase (DNA-PK), and MRE11, which are essential for the response to DNA damage and the recognition of exogenous DNA, also transmit their signals through the STING pathway [50,51]. Given that single-stranded and double-stranded RNA and DNA provoke innate immune reactions in vivo, this study is designed to assess the immunomodulatory potential of synthesized circDNA, circRNA, and commercially available 2′O-methylRNA inhibitors following in vivo administration. Following tail vein injection of the inhibitor, circDNA, and circRNA groups in mice, analysis via inflammatory factor array revealed that the inhibitor group elicited a more pronounced inflammatory response, with elevated levels of inflammatory factors and chemokines compared to the PBS control. In contrast, the circDNA and circRNA groups induced a milder inflammatory response than the inhibitor group. These findings suggest that circDNA and circRNA treatments may not trigger a robust immune response in vivo, potentially offering a safer therapeutic approach. However, this attenuated immune activation may also indicate a less potent therapeutic effect, warranting further investigation to optimize efficacy while maintaining safety.

Finally, our findings suggest potential clinical implications. Despite the significant challenges in developing circNAs-based therapeutics, several nucleic acid-based medications have been approved or are undergoing clinical trials. Designing circNAs to target and adsorb oncomiR functionality is crucial for expanding potential therapeutic target libraries and developing methods for treating diseases such as cancer.

## Figures and Tables

**Figure 1 biomedicines-13-01171-f001:**
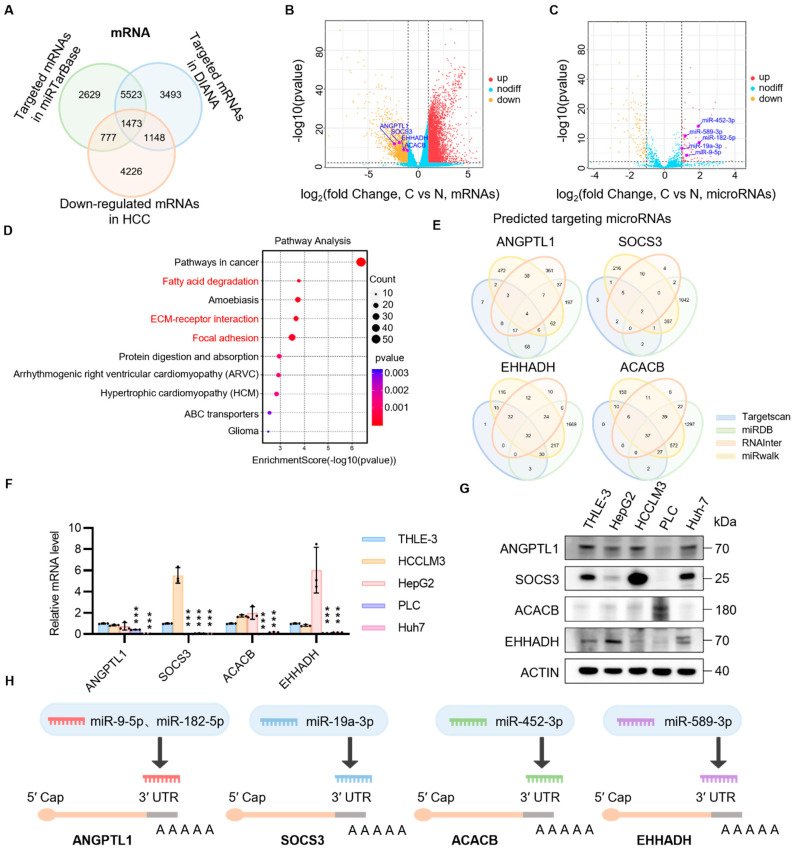
Screening for tumor suppressor genes targeted by OncomiRs in HCC. (**A**–**C**) Two databases were utilized to screen potential TSGs, which were then combined with TCGA database to identify ANGPTL1, SOCS3, ACACB, and EHHADH as promising candidates. (**D**) KEGG analysis between healthy subjects and hepatocellular carcinoma patients with hepatic cellular cancer were screened. (**E**) The oncomiRs of the candidate tumor suppressor gene were predicted using TCGA database and the miRDB, Target Scan, RNAInter, and miRwalk databases. (**F**) Quantitative polymerase chain reaction (qPCR) analysis of the mRNA expression of ANGPTL1, SOCS3, ACACB, and EHHADH in normal cells (THLE-3) and tumor cells (HepG2, HCCLM3, PLC, and Huh-7). Statistical significance was determined by Student’s *t*-test between normal cells and tumor cells. (**G**) Western blot analysis was conducted to assess the protein expression of ANGPTL1, SOCS3, ACACB, and EHHADH in normal cells (THLE-3) and HCC cells (HepG2, HCCLM3, PLC, and Huh-7). (**H**) Schematic diagram of circular nucleic acid sponge oncomiRs to rescue tumor suppressor gene expression. Statistical significance was determined by Student’s *t*-tests (*n* ≥ 3, *** *p* < 0.001).

**Figure 2 biomedicines-13-01171-f002:**
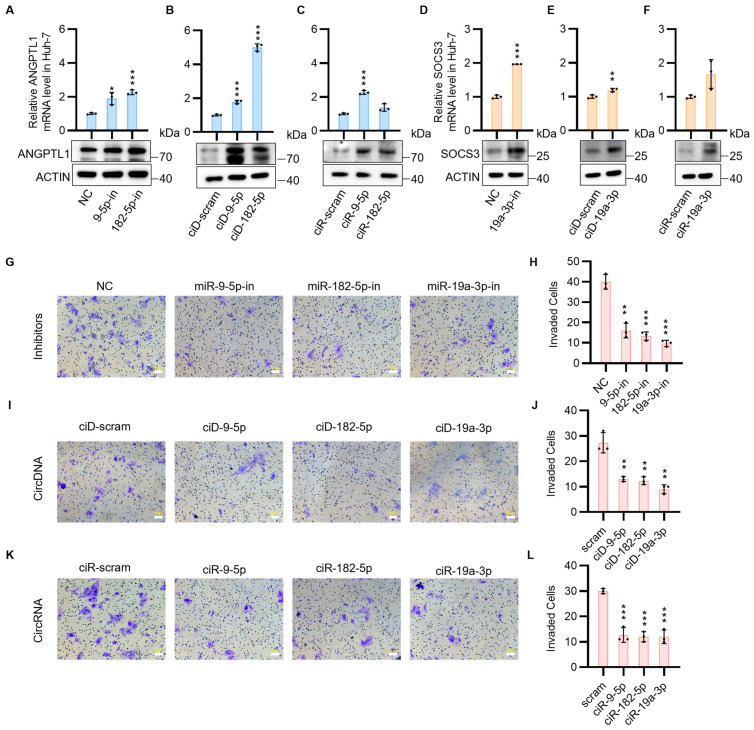
The expression of ANGPTL1 and SOCS3 is increased by circNAs, which consequently inhibit tumor migration. (**A**) ANGPTL1 mRNA levels (RT-qPCR) and protein levels (Western blot) after transfection with miRNA inhibitors (miR-9-5p-in, miR-182-5p-in) targeting oncomiRs. (**B**) ANGPTL1 mRNA levels (RT-qPCR) and protein levels (Western blot) post-transfection with circDNAs (circDNA-9-5p, circDNA-182-5p). (**C**) ANGPTL1 mRNA levels (RT-qPCR) and protein levels (Western blot) following circRNA transfection (circRNA-9-5p, circRNA-182-5p). (**D**) SOCS3 mRNA levels (RT-qPCR) and protein levels (Western blot) after miRNA inhibitor transfection (miR-19a-3p-in). (**E**) SOCS3 mRNA levels (RT-qPCR) and protein levels (Western blot) post-circDNA transfection (circDNA-19a-3p). (**F**) SOCS3 mRNA levels (RT-qPCR) and protein levels (Western blot) following circRNA transfection (circRNA-19a-3p). (**G**,**I**,**K**) Transwell migration assays of Huh-7 cells transfected with miRNA inhibitors (**G**: miR-9-5p-in, miR-182-5p-in; **I**: miR-19a-3p-in); circDNAs (**G**: circDNA-9-5p, circDNA-182-5p; **I**: circDNA-19a-3p); circRNAs (**K**: circRNA-9-5p, circRNA-182-5p, circRNA-19a-3p). Scale bars: 100 μm. (**H**,**J**,**L**) Quantification of migratory cells from (**G**,**I**,**K**), respectively. Statistical significance was determined by Student’s *t*-tests (*n* ≥ 3, * *p* < 0.05, ** *p* < 0.005, *** *p* < 0.001).

**Figure 3 biomedicines-13-01171-f003:**
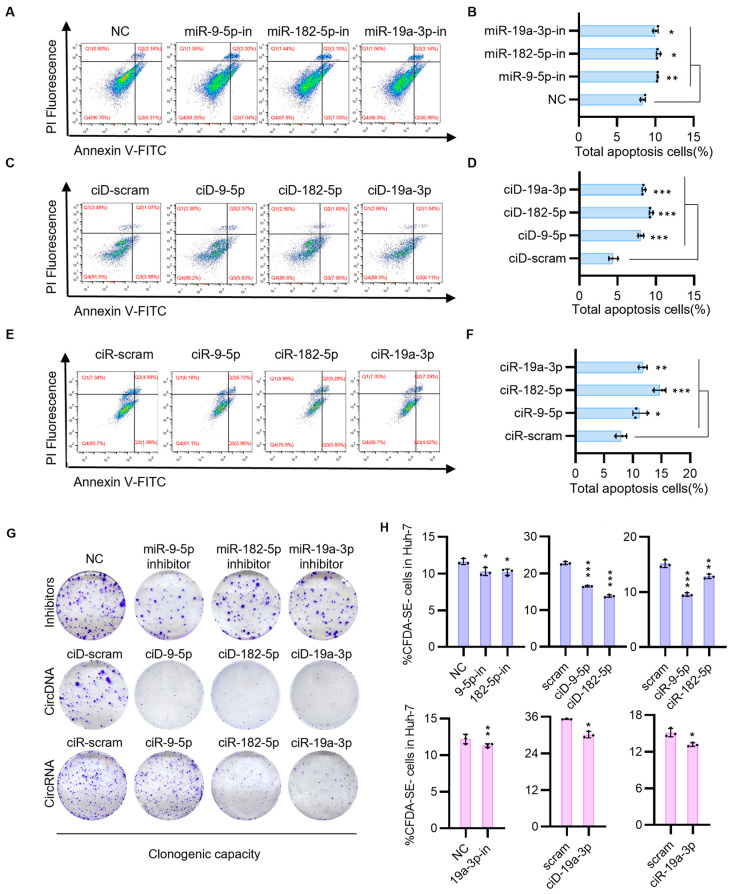
CircNAs transfection modulates tumor migration and apoptosis pathways to attenuate HCC progression. (**A**) The impact of miRNA inhibitors (miRNA-9-5p inhibitor, miRNA-182-5p inhibitor, miR-19a-3p inhibitor) on tumor cell apoptosis was evaluated using flow cytometry. Cells were first gated to exclude debris based on forward scatter (FSC) and side scatter (SSC), followed by selection of single cells using FSC-height (FSC-H) versus FSC-area (FSC-A) parameters. Apoptotic cells were quantified by Annexin V-FITC and propidium iodide (PI) staining. Total apoptotic cells were calculated as the sum of Annexin V^+^/PI^−^ (early apoptosis, Q2) and Annexin V^+^/PI^+^ (late apoptosis, Q3) populations. The proportion of apoptotic cells is shown. (**B**) A quantitative analysis of the flow cytometry data presented in (**A**) is provided below. (**C**) The impact of circDNAs (CircDNA-9-5p, CircDNA-182-5p, CircDNA-19a-3p) on tumor cell apoptosis was evaluated using flow cytometry, following identical gating procedures. (**D**) A quantitative analysis of the flow cytometry data presented in (**C**) is provided below. (**E**) The impact of circRNAs (CircRNA-9-5p, CircRNA-182-5p, CircRNA-19a-3p) on tumor cell apoptosis was evaluated using flow cytometry, following identical gating procedures. Red numeric annotations in panels (**A**,**C**,**E**) indicate the percentage of cells within each quadrant. (**F**) A quantitative analysis of the flow cytometry data presented in (**E**) is provided below. (**G**) To evaluate the proliferative impact of miRNA inhibitors, circDNAs, and circRNAs on Huh-7 cell growth, clone formation assays were conducted. (**H**) The CFDA SE fluorescent probe was utilized to label tumor cells and evaluate the proliferative impact of miRNA inhibitors, circDNAs, and circRNAs on tumor cell growth. Statistical significance was determined by Student’s *t*-tests (*n* ≥ 3, * *p* < 0.05, ** *p* < 0.005, *** *p* < 0.001).

**Figure 4 biomedicines-13-01171-f004:**
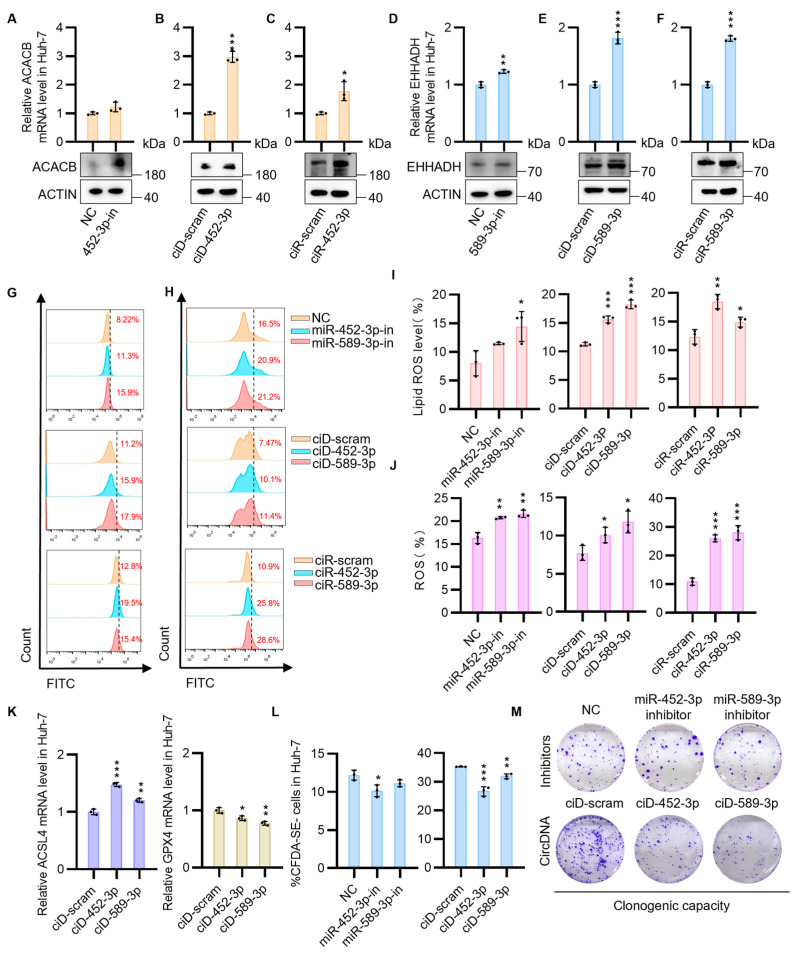
CircNAs facilitate ferroptosis and inhibit HCC progression by restoring ACACB and EHHADH expression. (**A**) RT-qPCR and Western blot analysis of ACACB mRNA and protein levels in Huh-7 cells transfected with miRNA-452-3p-in. (**B**) ACACB mRNA and protein expression in Huh-7 cells transfected with circDNA-452-3p. (**C**) ACACB mRNA and protein expression in Huh-7 cells transfected with circRNA-452-3p. (**D**) RT-qPCR and Western blot analysis of TSG EHHADH mRNA and protein levels in Huh-7 cells transfected with miRNA-589-3p-in. (**E**) EHHADH mRNA and protein expression in Huh-7 cells transfected with circDNA-589-3p. (**F**) EHHADH mRNA and protein expression in Huh-7 cells transfected with circRNA-589-3p. (**G**) BODIPY probe quantification of lipid peroxidation in Huh-7 cells transfected with miRNA inhibitors, circDNAs, and circRNAs. Cells were gated to exclude debris based on forward/side scatter (FSC/SSC), followed by single-cell selection using FSC-height (FSC-H) versus FSC-area (FSC-A). Fluorescence thresholds for lipid peroxidation were established using negative controls (inhibitor-nc, ciD-scram, ciR-scram). Fluorescence signals were acquired through the FITC channel (excitation/emission: 488 nm/525 ± 20 nm), with ROS-positive cells quantified as a percentage of the total population. (**I**) Quantitative analysis of flow cytometry results from (**G**) demonstrates the relative lipid ROS levels normalized to negative controls. (**H**) Total ROS levels in transfected Huh-7 cells were quantified using DCFH-DA with the same gating strategy as in (**G**) (FSC/SSC debris exclusion, FSC-H/FSC-A single-cell selection, FITC channel: 488/525 nm). (**J**) Quantitative analysis (**H**) shows ROS-positive cell percentages. (**K**) RT-qPCR quantification of ACSL4 and GPX4 mRNA expression levels in tumor cells transfected with circDNAs (CircDNA-452-3p, CircDNA-589-3p). (**L**) The CFDA SE fluorescent probe was used to label tumor cells and evaluate the proliferative effects of miRNA inhibitors and circDNAs on tumor cell growth. A quantitative analysis of the flow cytometry results is provided. (**M**) Clone formation assays were conducted to assess the impact of miRNA inhibitors and circDNAs on the proliferative capacity of Huh-7 cells. Statistical significance was determined by Student’s *t*-tests (*n* ≥ 3, * *p* < 0.05, ** *p* < 0.005, *** *p* < 0.001).

**Figure 5 biomedicines-13-01171-f005:**
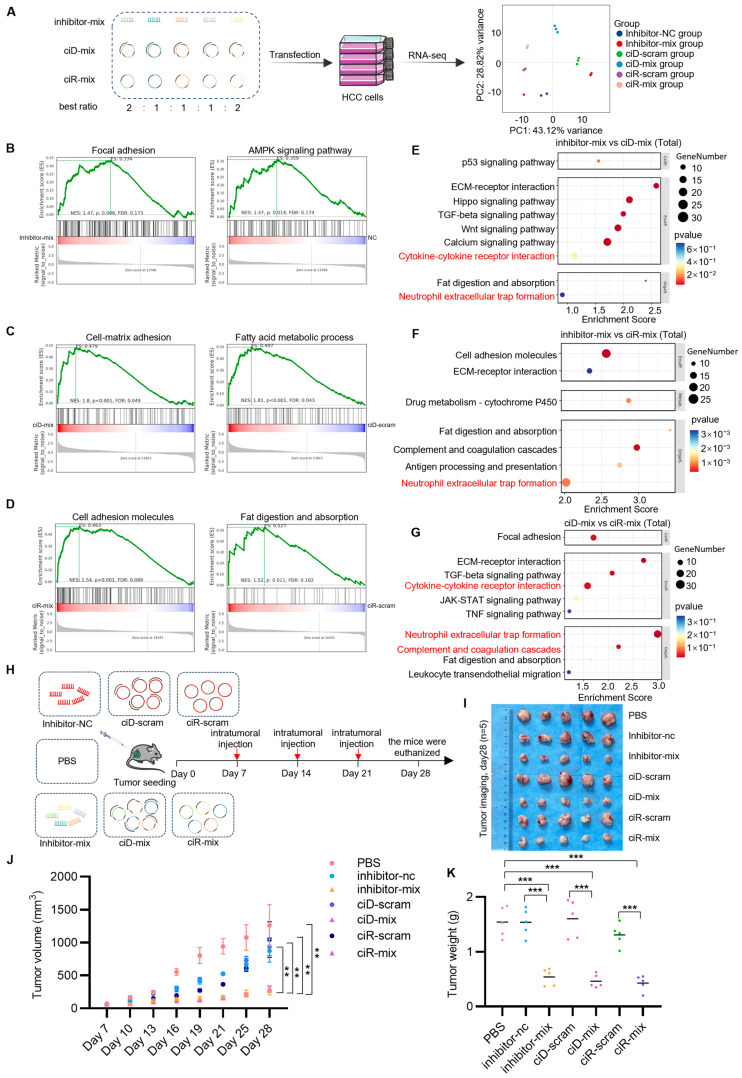
OncomiR sponge-mediated transcriptional rewiring drives durable HCC regression and immune reprogramming. (**A**) Transcriptomic profiling of HCC cells transfected with inhibitor, circDNA, or circRNA groups (schematic workflow shown). (**B**–**D**) GSEA of differentially expressed genes in Huh-7 cells transfected with (**B**) inhibitor-mix vs. nc, (**C**) ciD-mix vs. ciD-scram, (**D**) ciR-mix vs. ciR-scram (Normalized Enrichment Scores [NES] and q-values computed by GSEA). (**E**–**G**) RNA-seq-derived differential gene enrichment bubble plots comparing (**E**) inhibitor-mix vs. circDNA-mix, (**F**) inhibitor-mix vs. circRNA-mix, (**G**) circDNA-mix vs. circRNA-mix. (**H**) NOD/SCID mice (*n* = 5/group) were subcutaneously inoculated with Huh-7 cells (3 × 10^6^ cells/mouse). Starting on day 7 post-inoculation, intratumoral injections of ciD-mix, ciR-mix, inhibitor-mix, or controls (circDNA-scram, circRNA-scram, inhibitor-NC, PBS) were administered via jetPEI nanoparticles (N/P = 8) weekly for three doses. Tumor volumes were measured weekly. Experimental groups received intratumoral injections of inhibitor-mix (10 μg/mouse), circDNA (10 μg/mouse), or circRNA (10 μg/mouse), while control groups received equivalent volumes. Treatments were administered every 7 days for three cycles. (**I**) Tumor images of mice in each group after 21 days of circNAs treatment, with 7 groups and 5 mice per group. (**J**) Tumor volume growth curve of the inhibitor-mix group, ciD-mix group, and ciR-mix group after intratumoral injection, showing changes in tumor volume during the study. (**K**) Changes in tumor weight after intratumoral injection of inhibitor-mix, ciD-mix, and ciR-mix groups. Each color represents different groups of tumor individual weights. Color coding: Red: PBS group; Blue: inhibitor-nc group; Yellow: inhibitor-mix group; Pink: ciD-scram group; Purple: ciD-mix group; Green: ciR-scram group; Violet: ciR-mix group. Statistical significance was determined by Student’s *t*-test (*n* = 3, ** *p* < 0.01, *** *p* < 0.001).

**Figure 6 biomedicines-13-01171-f006:**
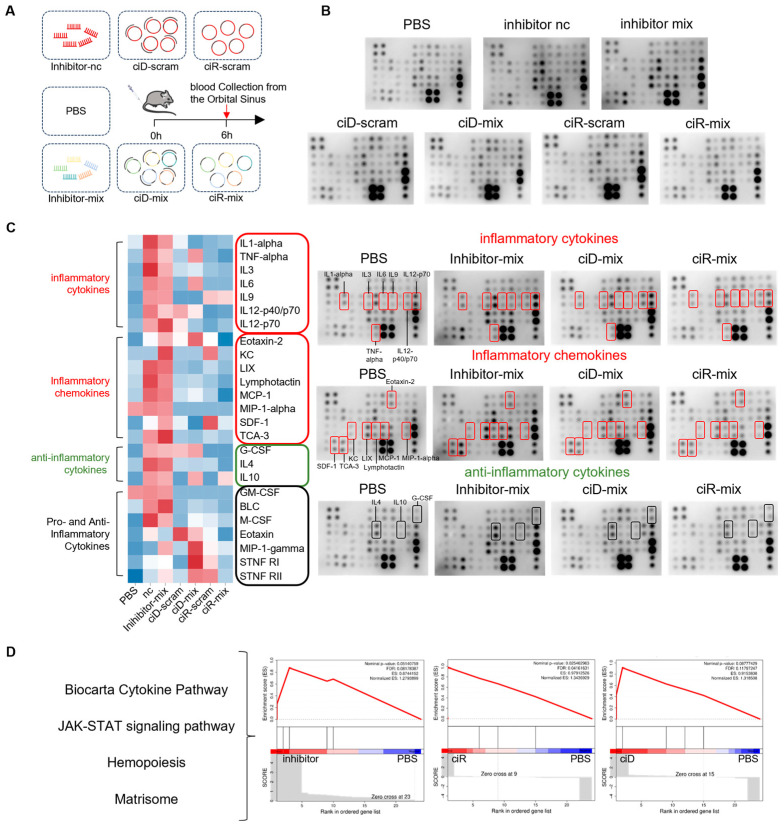
OncomiRs sponge vectors elicit inflammatory factor activation. (**A**) Schematic representation of the experimental protocol involving the administration of seven distinct treatment groups to 6-week-old female BALB/c mice, followed by orbital cavity blood collection 6 h post-injection. Each group consisted of three replicates, totaling seven groups. (**B**) Analysis of serum inflammatory factors using an antibody array following retro-orbital blood collection. (**C**) Heatmap analysis of the inflammation array, categorizing the density of inflammatory factors into pro-inflammatory cytokines, pro-inflammatory chemokines, anti-inflammatory agents, and pro- and anti-inflammatory cytokines. Color-coded bounding boxes indicate functional classifications: red (pro-inflammatory cytokines & chemokines), green (anti-inflammatory agents), and black (dual-role cytokines with pro-/anti-inflammatory properties). (**D**) Cellular pathway enrichment analysis to elucidate the differential profiles of inflammatory factors across treatment groups.

**Figure 7 biomedicines-13-01171-f007:**
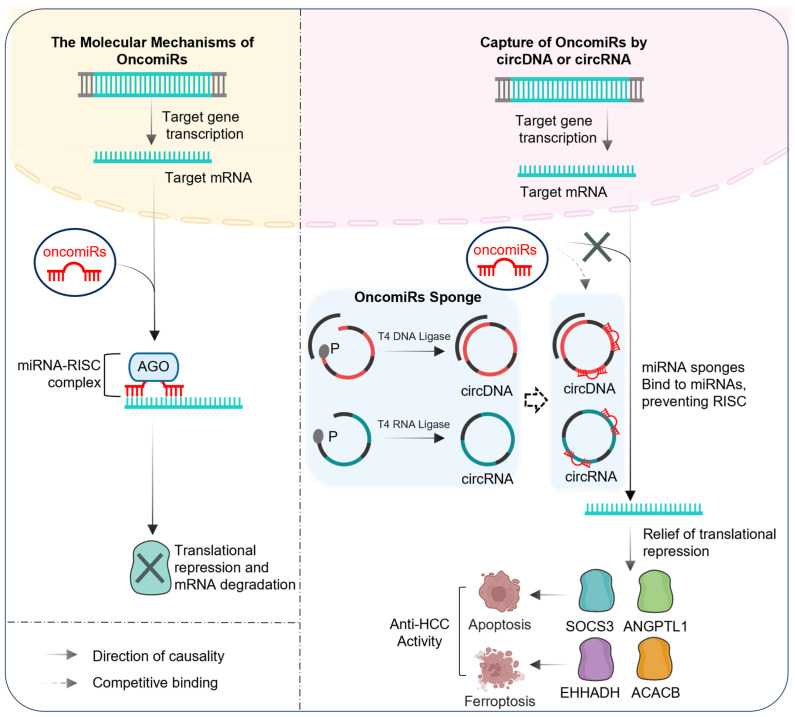
Design and mechanistic diagram of anti-tumor activity of circNAs.

## Data Availability

Data are contained within the article and Appendix A.

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
