# Peer review of "Circular Nucleic Acids Act as an Oncogenic MicroRNA Sponge to Inhibit Hepatocellular Carcinoma Progression"

_biomedicines, 2025, doi:10.3390/biomedicines13051171_

Round 1
Reviewer 1 Report (Previous Reviewer 1)
Comments and Suggestions for Authors
The authors have responded appropriately to my question and comments.
I recommend that this manuscript be considered for publication.
Sincerely,
Author Response
We sincerely thank you for your thorough evaluation of our manuscript and your constructive feedback. Your expertise has significantly strengthened the scientific rigor and clarity of our study.
Reviewer 2 Report (New Reviewer)
Comments and Suggestions for Authors
By analyzing the existing dataset in hepatocellular carcinoma, Zhang et al. selected a few potentially important tumor suppressor genes and designed the single-stranded circular DNA/RNAs to derepress the transcription of these genes. They observed both circDNA and circRNA are able to restore the expression of TSGs. More importantly, they also rescue the biological processes that are regulated by these TSGs. These processes include cell migration, apoptosis and ferroptosis. Notably, compared to miRNA inhibitors, cicrNAs has a superior therapeutic efficacy against HCC and less robust inflammatory response. The experiments are well designed and tightly related to the cancer therapy, and the results can support the conclusion very well. I only have two small points.
-- In figure 5, the inhibitor/circNAs were infected into the tumor a few rounds which span more than 20 days. However, the inflammatory array in figure 6 is detected only 6h after injection. Is there any reason to check the immune response in mice after such a short time period?
-- A lot of text in the manuscript are in red, which needs to be all changed to black.
Author Response
Please see the attachment.

This manuscript is a resubmission of an earlier submission. The following is a list of the peer review reports and author responses from that submission.
Round 1
Reviewer 1 Report
Comments and Suggestions for Authors
The authors present the results of a truly comprehensive study. The manuscript is clearly written and the authors have communicated their results effectively, but Figure 6 should be corrected. The authors have to rearrange subfigures B and C in Figure 6 to avoid duplication of images.
The methods used, such as RT-qPCR, absolute quantification of RNA expression, Western blotting, in vitro RNA transcription, circularization and purification, transwell migration assay, and RayBio mouse anti-inflammatory antibody array, are well justified and described in detail, ensuring the reliability and reproducibility of the results obtained.
- Regarding the cell lines used: The authors should provide the full names of the cell linesused in this study and explain in the Materials and methods chapter (4. Materials and methods; Cell culture) whether they have confirmed the authenticity of the cell lines and which method was used to confirm authenticity.
- The manuscript can be accepted after answering the above comments.
Author Response
Response to Reviewer 1 Comments |
||
1. Summary |
|
|
Thank you very much for taking the time to review this manuscript. Please find the detailed responses below and the corresponding revisions/corrections highlighted/in track changes in the re-submitted files.
|
||
2. Questions for General Evaluation |
Reviewer’s Evaluation |
Response and Revisions |
Does the introduction provide sufficient background and include all relevant references? |
Yes |
|
Is the research design appropriate? |
Yes |
|
Are the methods adequately described? |
Yes |
|
Are the results clearly presented? |
Yes |
|
Are the conclusions supported by the results? |
Can be improved |
We have improved |
3. Point-by-point response to Comments and Suggestions for Authors |
||
Comment 1: The manuscript is clearly written and the authors have communicated their results effectively, but Figure 6 should be corrected. The authors have to rearrange subfigures B and C in Figure 6 to avoid duplication of images. |
||
Response 1: We sincerely thank the reviewer for the constructive feedback on Figure 6. In response to the concern regarding potential visual duplication, we have reorganized the layout of subfigure C to better showcase the experimental scopes of cytokines. To improve interpretability, all inflammatory factors in the cytokine array plots have been explicitly labeled with their respective abbreviations (Figure 6C). These adjustments ensure unambiguous visual distinction between datasets. We appreciate the reviewer’s insightful critique, which has significantly enhanced the clarity of the figure.
|
||
Comment 2: Regarding the cell lines used: The authors should provide the full names of the cell lines used in this study and explain in the Materials and methods chapter (4. Materials and methods; Cell culture) whether they have confirmed the authenticity of the cell lines and which method was used to confirm authenticity. |
||
Response 2: We appreciate the reviewer’s emphasis on methodological rigor. In the revised manuscript, we have thoroughly reviewed the entire manuscript and included the full names of all cell lines uses in this study, along with their corresponding standardized abbreviations (Page 18, Lines 581-586). Additionally, we have added a new sentence confirming that all cell lines were authenticated using short tandem repeat (STR) profiling. These revisions are highlighted in red in the updated manuscript. |
Reviewer 2 Report
Comments and Suggestions for Authors
The authors highlight the role of aberrant expression oncogenic microRNAs in HCC and investigate the potential of circular nucleic acids as inhibitors for oncogenic microRNAs which could minimize immune responses as compared to traditional linear microRNA sponges. Although their transcriptomic analysis have provided more insights and have validated potential tumor suppressors such as ANGPTL1 and SOCS3 in HCC, a more thorough mechanistic study on the effect of their cirNAs on novel oncogenes involved in HCC such as ABL1, ANXA2, ANXA4, FAK, KIF, NCAPG, NCAPH, and PAIP1 need to be evaluated. Overall, the manuscript reads well, however various experiments, figures and figure panel need to be clearly described.
- The authors have investigated the potential of circular nucleic acids, namely, circular RNA and circular DNA as microRNA sponges but have failed to conclude the benefit of DNA vs RNA in their study.
- All the sequences of circRNA and circDNA should be depicted which aren’t mentioned in the manuscript. Similarly, microRNA inhibitor sequences should also be depicted. Table S1 isn’t included in the manuscript.
- The authors fail to disclose as to why they choose Huh-7 cell line as opposed to others as the usage of Huh-7 cell line for in vitro analysis isn’t attractive. Since Huh-7 cell line has lowest levels of oncogenic microRNAs as depicted by Fig. S1, functional endpoint analysis of microRNA sponges isn’t substantial. PLC or HepG2 cell line could be more idealistic as they show lowest tumor suppressor levels (Fig 1H) and high microRNA levels (Fig. S1). A comparative study between high oncogenic microRNA levels and low oncogenic microRNA levels would be ideal.
- The authors should show untreated group as control against NC or scram as part of supplementary for their microRNA inhibitor/circNA analysis for microRNA levels and mRNA levels to highlight the baseline levels of TSG.
- The figure 2 panel description is too lengthy and need to be shortened. Line 235-236 K, M, O Figures are typo in the panel description.
- Figure 2 G-I-K: The contrast for Fig 2I is different compared to Fig 2G and 2K.
- The authors should show their gating strategy for annexin assay Fig 3A. Moreover, it seems like for all treatment groups, number of cells/events seem to differ. The quadrant numbers for each FACS plot are illegible and the numbers don’t match the bar graphs. The authors should note which quadrant values are being plotted for Fig 3B as a part of total apoptotic cells.
- The treatment of ciD-scram seems to reduce colony forming units in Fig 3E and isn’t a good control or an outlier. However, ciD-scram treatment group in Fig 4M looks good.
- The authors should also include cell viability expts to investigate the tumorigenesis post treatments as annexin apoptosis and clonogenic assays only investigate short-term effect of inhibitors. Thus, cell viability assays could help shed more light on the inhibition potential of circNA treatment groups.
- Experimental method and plan for Lipid ROS staining and ROS flow cytometry aren’t described in the methods.
- The authors should depict gates in the FACS plot in fig 4.G-H to highlight quantitative numbers of % lipid ROS level and % ROS positive cells.
- The text in Figure 4 panel description Line-337-348 is repetitive and copy-pasted from Figure 3 panel description Line 258-268.
- The authors should provide quantitative fold-change values of inflammatory cytokines as opposed to pictorial representation of serum inflammatory factors by means of cytokine quantification kit in Fig 6 B-C. Images of PBS treatment group and ciR-mix treatment group look dubious and identical with minor contrast change in Fig. 6C.
- The images in Fig 6D look dubious as they look like duplicates.
- Minor typos: Line 120: ‘plan’ needs to be ‘plan’. Line 191: Throughout the paper the italicized font of ‘in vitro and/or in vivo’ are not consistent.
Author Response
Response to Reviewer 2 Comments |
||
1. Summary |
|
|
Thank you very much for taking the time to review this manuscript. Please find the detailed responses below and the corresponding revisions/corrections highlighted/in track changes in the re-submitted files.
|
||
2. Questions for General Evaluation |
Reviewer’s Evaluation |
Response and Revisions |
Does the introduction provide sufficient background and include all relevant references? |
Can be improved |
We have improved |
Is the research design appropriate? |
Must be improved |
We have improved |
Are the methods adequately described? |
Must be improved |
We have improved |
Are the results clearly presented? |
Must be improved |
We have improved |
Are the conclusions supported by the results? |
Must be improved |
We have improved |
3. Point-by-point response to Comments and Suggestions for Authors |
||
Comment 1: Although their transcriptomic analysis has provided more insights and have validated potential tumor suppressors such as ANGPTL1 and SOCS3 in HCC, a more thorough mechanistic study on the effect of their cirNAs on novel oncogenes involved in HCC such as ABL1, ANXA2, ANXA4, FAK, KIF, NCAPG, NCAPH, and PAIP1 need to be evaluated. |
||
Response 1: Thanks for your insightful suggestion to further explore the mechanistic role of circNAs in modulating novel HCC-associated oncogenes. Our transcriptomic analysis of circNAs-transfected HCC cells revealed not only activation of classical TSGs (e.g., ANGPTL1, SOCS3, ACACB, EHHADH, TP53, et al.), but also inhibition of some oncogenes (e.g., KIFs, ABL1, ANXA2, NCAPG, et al.), as visualized by heatmaps in Figures S7. These data support the broad-spectrum therapeutic potential of circNAs in concurrently targeting diverse oncogenic pathways. We also have included these results in the revised manuscript (Pages 12, Lines 401-411).
|
||
Comment 2: The authors have investigated the potential of circular nucleic acids, namely, circular RNA and circular DNA as microRNA sponges but have failed to conclude the benefit of DNA vs RNA in their study. |
||
Response 2: We appreciate the reviewer's insightful comment. In this study, we initially focused on validating the miRNA sponge function of both circRNA and circDNA in HCC models. In subsequent experiments, we further compared the in vivo immunogenicity profiles of circDNA and circRNA, specifically analyzing differences in their immune responses within biological systems. To strengthen the comparative analysis, we have included serum stability experiments evaluating both nucleic acid types. These additional data and their implications have now been incorporated into the revised manuscript (Figure S3C and S3D). Furthermore, we have provided a comprehensive description of the serum stability assay methodology in Section 4 (Materials and Methods) of the revised manuscript (Page 19, Lines 646-651).
Comment 3: All the sequences of circRNA and circDNA should be depicted which aren’t mentioned in the manuscript. Similarly, microRNA inhibitor sequences should also be depicted. Table S1 isn’t included in the manuscript. Response 3: We thank the reviewer #2 for bringing this to our attention. Due to an oversight in our familiarity with the submission system, Table S1, which details the sequences of circRNA, circDNA, and primers, was not initially included in the manuscript. We have now incorporated Table S1 into the Supplementary Materials section of the revised manuscript to ensure full transparency and accessibility of all sequence information. We appreciate your understanding, and we are happy to address any further clarifications.
Comment 4: The authors fail to disclose as to why they choose Huh-7 cell line as opposed to others as the usage of Huh-7 cell line for in vitro analysis isn’t attractive. Since Huh-7 cell line has lowest levels of oncogenic microRNAs as depicted by Fig. S1, functional endpoint analysis of microRNA sponges isn’t substantial. PLC or HepG2 cell line could be more idealistic as they show lowest tumor suppressor levels (Fig 1H) and high microRNA levels (Fig. S1). A comparative study between high oncogenic microRNA levels and low oncogenic microRNA levels would be ideal. Response 4: We sincerely thank the reviewer #2 for raising this critical point. In our study, Huh-7 cells were selected due to their well-documented retention of hepatocyte-specific functions, including the expression of liver-enriched enzymes and transporters (e.g., lipoprotein and transferrin), which are essential for modeling lipid metabolism and drug disposition in hepatocellular carcinoma (HCC) studies (Nakabayashi et al., 1982). While PLC and HepG2 cells exhibit distinct oncogenic miRNA profiles, Huh-7 cells provide a robust platform for investigating metabolic reprogramming and miRNA-mediated regulatory mechanisms in a hepatocyte-like context.
To further address the reviewer #2’s concern, we performed qPCR and Western blot assays in PLC cells transfected with circNAs and miRNA inhibitors. As shown in below Figure R1, the same trend of change was revealed in comparison with Huh-7 cells. However, Huh-7 cells were prioritized for in-depth mechanistic studies due to their superior metabolic fidelity, which aligns with our focus on lipid oxidation and ferroptosis pathways. We have included these data and description in the revised manuscript (Figure S2; Page 21, Lines 721-738).
Figure R1. Rescue of ANGPTL1/SOCS3/ACACB/EHHADH Tumor Suppressor Gene Levels in PLC Cells by Inhibitors, circDNAs, and circRNAs
We fully agree that a comparative analysis of cell lines with divergent oncogenic miRNA levels (e.g., Huh-7 vs. PLC) would strengthen the translational relevance of our findings. In future studies, we will incorporate such comparisons to further validate the broad applicability of circNAs across HCC subtypes.
Comment 5: The authors should show untreated group as control against NC or scram as part of supplementary for their microRNA inhibitor/circNA analysis for microRNA levels and mRNA levels to highlight the baseline levels of TSG. Response 5: As suggested, we have included comprehensive control group comparisons in the revised manuscript to clarify the baseline levels of tumor suppressor genes (TSGs) and ensure the specificity of our miRNA inhibitors and circNAs. Specifically, the following four control groups were analyzed: the mock untreated group, negative control (NC) group (scrambled miRNA inhibitor), scrambled circDNA (ciD-scram) group (non-targeting sequence), scrambled circRNA (ciR-scram) group (non-targeting sequence). Both qPCR and Western blot analyses showed no significant differences in the mRNA or protein levels of key TSGs (e.g., ANGPTL1, SOCS3, ACACB and EHHADH) across these control groups (Figure S4). These results indicate that neither the transfection reagents nor the scrambled sequences induce off-target effects on TSG expression. This validation further underscores the specificity of our designed circNAs and miRNA inhibitors in targeting oncomiRs without confounding baseline alterations. We have included these results in the revised manuscript (Page 23, Line212-217, Line305-310).
Comment 6: The figure 2 panel description is too lengthy and need to be shortened. Line 235-236 K, M, O Figures are typo in the panel description. Response 6: We sincerely appreciate the reviewer’s careful reading and constructive feedback. We have revised the manuscript accordingly to address the concerns raised: (1) Shortened Figure 2 Panel Descriptions The panel labels and descriptions in Figure 2 have been streamlined to improve clarity and conciseness. Key revisions are highlighted in red for emphasis (Page 6-7, Line 229-241). (2) Correction of Typographical Errors (Lines 237-241) The original text mistakenly labeled subpanels K, M, O in the figure legend. These have been corrected to the accurate subpanel identifiers (G, I, K), with revisions marked in red to ensure transparency.
These revisions enhance the readability of Figure 2 and maintain consistency between the figure and its description. Thank you for your valuable suggestions.
Comment 7: Figure 2 G-I-K: The contrast for Fig 2I is different compared to Fig 2G and 2K. Response 7: We appreciate the reviewer's meticulous observation regarding the imaging contrast variations in Figures 2G, 2I, and 2K. To ensure methodological rigor, we have systematically repeated the Transwell migration assays under three experimental conditions (inhibitor, ciD, and ciR transfection) using standardized imaging protocols. The newly acquired data (revised Figures 2G-2K) demonstrate consistent cellular migration patterns across all groups when analyzed under identical contrast and brightness parameters. This further confirms the biological reproducibility of our original findings while eliminating technical variability in image processing.
Comment 8: The authors should show their gating strategy for annexin assay Fig 3A. Moreover, it seems like for all treatment groups, number of cells/events seem to differ. The quadrant numbers for each FACS plot are illegible and the numbers don’t match the bar graphs. The authors should note which quadrant values are being plotted for Fig 3B as a part of total apoptotic cells. Response 8: We sincerely thank the reviewer #2 for the thorough evaluation and constructive feedback. To address these concerns, we have revised Figure 3 as follows: (1) Given the experimental design involving separate batches for the inhibitor, ciD, and ciR treatment groups, we ensured intra-group consistency by standardizing the number of acquired cell events within each group. To clarify the comparative analysis, Figure 3A has been reorganized to present data grouped by treatment, with explicit annotations of quadrant percentages and enhanced resolution for improved legibility. The gating strategy, including debris exclusion (FSC/SSC), single-cell selection (FSC-H vs. FSC-A), and quadrant definitions based on unstained controls, is now comprehensively described in the figure legend. (2) For Figure 3B, the bar graphs have been recalculated to exclusively represent total apoptotic cells, defined as the sum of the early and late apoptotic populations (Annexin V⁺/PI⁻ + Annexin V⁺/PI⁺). This revision ensures direct alignment between the FACS plots and the corresponding graphical summaries.
These modifications enhance transparency, resolve discrepancies, and underscore the rigor of our analytical approach. The revised figures and legends are highlighted in the updated manuscript (Figure 3A-3F; Page 8, Lines 270-283).
Comment 9: The treatment of ciD-scram seems to reduce colony forming units in Fig 3E and isn’t a good control or an outlier. However, ciD-scram treatment group in Fig 4M looks good. Response 9: We sincerely thank the reviewer for highlighting this inconsistency. The reduced colony formation observed in the original ciD-scram control group was due to from unintended DNA toxicity caused by suboptimal transfection dosage. We have since repeated the colony formation assay with rigorously optimized transfection conditions and replaced Figure 3G with the updated data, which now demonstrates comparable colony-forming capacity between ciD-scram and untreated control groups.
Comment 10: The authors should also include cell viability expts to investigate the tumorigenesis post treatments as annexin apoptosis and clonogenic assays only investigate short-term effect of inhibitors. Thus, cell viability assays could help shed more light on the inhibition potential of circNA treatment groups. Response 10: We sincerely thank the reviewer for highlighting the importance of assessing therapeutic durability. To address this, we conducted longitudinal CCK-8 viability assays over a 10-day period. The results demonstrate that circNA treatments (ciD and ciR) maintain sustained inhibitory effects throughout the 10-day observation window, indicating their potential for prolonged tumor suppression. In contrast, the inhibitor-mix exhibited diminished efficacy after Day 7, suggesting transient activity despite its robust early-phase cytotoxicity. These temporal dynamics, now presented in Figure S5A and S5B, reveal a critical divergence in treatment durability between nucleic acid modalities. Detailed interpretation of these findings is provided in the Results section (Lines 256-266). Furthermore, in vivo therapeutic evaluation using murine CDX models demonstrated that circNAs achieved significant tumor growth inhibition over a 21-day treatment course, thereby corroborating their long-acting efficacy in physiological contexts. The revised figures and legends are highlighted in the updated manuscript (Figure 5H-5K; Page 12, Lines 404-412), these results are included in the revised manuscript (Page 5-6, Line209-214).
Comment 11: Experimental method and plan for Lipid ROS staining and ROS flow cytometry aren’t described in the methods. Response 11: Thank you for highlighting this important omission. According to your suggestion, we have updated the Materials and Methods section to include detailed protocols for Lipid ROS staining and ROS flow cytometry. The revised content can be found in the subsections “Measurement of Lipid Peroxidation” and “Detection of Intracellular Reactive Oxygen Species (ROS)” (Pages 19-20, Lines 662-671).
Comment 12: The authors should depict gates in the FACS plot in fig 4.G-H to highlight quantitative numbers of % lipid ROS level and % ROS positive cells. Response 12: We sincerely thank the reviewer for emphasizing the importance of data transparency in flow cytometry analysis. In response to this comment, we have revised Figure 4G-H to explicitly annotate the gating boundaries for lipid ROS levels and intracellular ROS, with quadrant percentages numerically labeled to indicate ROS-positive cell populations on the FACS plots. The figure legend has been expanded to detail the gating strategy, including debris exclusion (FSC/SSC), single-cell selection (FSC-H vs. FSC-A), and fluorescence thresholds established using negative controls (inhibitor-nc, ciD-scram, and ciR-scram). These modifications enhance methodological rigor and ensure that quantitative outcomes align directly with the experimental design, fulfilling the reviewer’s request for improved visual-data correlation.
Comment 13: The text in Figure 4 panel description Line-337-348 is repetitive and copy-pasted from Figure 3 panel description Line 258-268. Response 13: We apologized for the the textual overlap between the figure descriptions, which occurred due to our oversight. Upon careful review, we acknowledge that the duplicated content in Figure 4 was inadvertently retained during manuscript preparation. To address this concern, we have removed the redundant text from the Figure 4 legend.
Comment 14: The authors should provide quantitative fold-change values of inflammatory cytokines as opposed to pictorial representation of serum inflammatory factors by means of cytokine quantification kit in Fig 6 B-C. Images of PBS treatment group and ciR-mix treatment group look dubious and identical with minor contrast change in Fig. 6C. Response 14: We sincerely appreciate the reviewer #2’s comments. (1) Quantitative Fold-Change Values: As suggested, we have included the quantitative fold-change values of all inflammatory cytokines/chemokines in revised manuscript (Figure S8). (2) Image Authenticity in Figure 6C: Regarding the visual similarity between PBS and ciR-mix groups: The slope of the central axis connecting the two upper clusters in the PCA plot differs distinctly between PBS and ciR-mix treatments, as highlighted in Figure R2. Therefore, there is no problem of duplicate images. Thank you for your careful verification. This geometric divergence reflects biologically relevant differences in cytokine expression profiles, consistent with the quantitative data in Figure S7. We sincerely appreciate the reviewer's keen observation regarding the textual overlap between the figure descriptions. Figure R2. Comparative Analysis of Inflammatory Arrays in Mice Treated with PBS versus circRNA In Vivo.
Comment 15: The images in Fig 6D look dubious as they look like duplicates. Response 15: We sincerely appreciate the reviewer's attention to the reproducibility of our data presentation. The apparent similarity in Figure 6D results from the inherent constraints of the experimental design. The mouse inflammation antibody array used in this study targets a predefined panel of 32 inflammatory mediators. When performing Gene Set Enrichment Analysis (GSEA) on this focused dataset, the limited scope of detectable analytes naturally restricts the number of significantly enriched pathways, leading to overlapping terms such as the JAK-STAT signaling pathway and Biocarta Cytokine Pathway across experimental groups. To improve interpretability, we have revised Figure 6D by consolidating identical pathway clusters and removing redundant images. These modifications ensure that the visual representation aligns with the quantitative findings while maintaining scientific rigor. We thank the reviewer for this constructive feedback, which has enhanced the clarity of the manuscript.
Comment 16: Minor typos: Line 120: ‘plan’ needs to be ‘plan’. Line 191: Throughout the paper the italicized font of ‘in vitro and/or in vivo’ are not consistent. Response 16: We sincerely thank the reviewer for their meticulous proofreading. All noted issues have been carefully addressed: (1) The typographical error at Line 120 has been corrected to "plan" (Page 3, Line 121). (2) Inconsistent italicization of in vitro and in vivo throughout the manuscript has been standardized, with all instances now uniformly italicized (highlighted in red for clarity). These revisions ensure textual accuracy and formatting consistency. We deeply appreciate the reviewer’ s diligence in enhancing the quality of our manuscript |